# Carbazole and Diketopyrrolopyrrole-Based D-A π-Conjugated Oligomers Accessed via Direct C–H Arylation for Opto-Electronic Property and Performance Study

**DOI:** 10.3390/molecules27249031

**Published:** 2022-12-18

**Authors:** Xiafeng Zhang, Lingwei Feng, Kai Zhang, Shi-Yong Liu

**Affiliations:** 1Jiangxi Provincial Key Laboratory of Functional Molecular Materials Chemistry, College of Materials, Metallurgical and Chemistry, Jiangxi University of Science and Technology, Ganzhou 341000, China; 2School of Materials Science and Engineering, South China University of Technology, Guangzhou 510000, China

**Keywords:** pi-conjugated oligomers, C–H direct arylation, organic photovoltaics, diketopyrrolopyrrole, carbazole

## Abstract

Five carbazole and diketopyrrolopyrrole-based donor-acceptor (D-A) new π-conjugated oligomers (π-COs) with gradually elongated lengths are facilely synthesized via a single pot of direct C–H arylation with merits of atom- and step-economy. The structure-property-performance correlations of these π-COs and their parent polymer are studied in detail by opto-electronic characterizations and bulk heterojunction (BHJ) organic photovoltaic (OPV) devices. It is found that the π-COs having longer lengths enable better performance in OPVs owing to the enhanced intermolecular interaction with the elongation of the conjugations. The above results not only highlight the powerful synthetic strategy here provided, but also reveal that π-COs with unique properties might find promising application in OPVs.

## 1. Introduction

Bulk heterojunction (BHJ) organic photovoltaic (OPV) has received considerable attention in recent years due to its outstanding merits of low cost, flexibility, lightweight, and solution processibility etc. The rapid progress of donor & acceptor materials and the optimization of devices has enabled the power conversion efficiency (PCE) of OPV to attain a new stage and thus highlights the great potential of π-functional materials for device applications [1,2,3,4,5,6,7,8,9,10,11,12,13,14,15]. Although OPVs based on π-polymers have been rapidly developed, there are still some inherent shortcomings, such as poor reproducibility and the uncertain molecular weight of polymers. π-conjugated oligomers (π-COs), as bridges between small molecules and polymers, having defined structures, high MWs, high crystallinity and orderly packing characters, are expected to combine the advantages of small molecules and polymers, while overcoming their respective shortcomings [4,16,17,18,19,20,21,22,23,24,25,26,27,28]. Meanwhile, π-COs with gradually increasing chain lengths can help to establish the distinct structure-property-performance correlations and provide guidance for the design of OPV materials [16,17,18,19,20,21,22].

Despite the abovementioned potentials, the synthetic tools for efficient access of π-COs, especially for the long-chain ones, are still very limited. Until now, approaches to π-conjugated materials mainly rely on the Pd-catalyzed C–M/C–X (M = B or Sn, X = Br or I) Suzuki or Stille cross couplings involving the pre-functionalization of precursors with C–M bonds under harsh conditions, such as anhydrous strong bases, and low reaction temperatures [29]. Meanwhile, the poor compatibility of active C^δ−^–M^δ+^ bonds with the electrophilic groups will be unfavorable to the structural diversity, and thus the controllable and scalable synthesis of π-functional materials. It is known that C–H represent the most widely distributed bond among the organic compounds. The direct transformation of C–H bonds [30,31] to the desired functional groups (i.e., direct functionalization of C–H bonds) is thus considered as the preferential method to shorten synthetic steps, reduce side-products, and improve the atom economy. In recent years, direct C–H arylation (i.e., C–H/C–X couplings) has emerged as one of the most promising alternatives to the classical C–M/C–X (M = B or Sn) cross couplings for the synthesis of π-conjugated materials, owing to its outstanding merits of functional group compatibility, atom economy, and cost-effectiveness [32,33,34,35,36,37,38,39,40].

In this study, a series of carbazole (Cz) and diketopyrrolopyrrole (DPP)-based donor-acceptor (D-A) novel homologous π-COs with gradually increasing repeat units (i.e., DPP-Cz-DPP (**O**1), DPP-(Cz-DPP)_2_-DPP (**O**2), DPP-(Cz-DPP)_3_-DPP (**O**3), DPP-(Cz-DPP)_4_-DPP (**O**4) and DPP-(Cz-DPP)_5_-DPP (**O**5)), accordingly with 3, 5, 7, 9 and 11 monomers, and molecular weights ranging between 1449~5161, were simultaneously produced in a single pot via direct C–H arylation reaction (Figure 1), and facilely separated by a single chromatographic column (CC), affording acceptable yields. The structure-property-performance correlations of these five π-COs with gradually increasing lengths were systematically studied by opto-electrochemical tests and OPV devices, which provides a guidance for the design of π-COs-based OPV materials.

## 2. Results and Discussion

### 2.1. Synthesis and Characterizations

The synthetic routes of the five DPP-Cz-based D-A π-COs and their parent polymer are depicted in Figure 1. DPP and Cz were chosen, respectively, as electron-accepting and -donating units owing to their outstanding merits of tunable opto-electronic properties and high charge mobility. Carbazole, as one of the most popular electron-rich monomers in the field of organic optoelectronics, has been widely studied due to its special electricity, electrochemical and photophysical properties and thermal stability over the past 20 years. The *p*-π conjugation of the lone paired electrons on the N atom endows Cz with a distinct electron-donating ability and tunable highest occupied molecular orbital (HOMO) levels, which is beneficial to modulating open circuit voltage (*V*_OC_) for OPVs. DPP-Cz-based D-A polymers have been employed as donor materials in OPVs [41,42,43,44]. Typically, the D-A architecture of π-materials will induce intramolecular charge transfer (ICT) and allow light absorption shift to longer wavelengths. Here, five DPP-Cz-based A.A-D-A.A backboned π-COs, i.e., **O**1, **O**2, **O**3, **O**4, **O**5 and their parent polymer **P**1 are designed and synthesized for a structure-property-performance correlation study.

As shown in Figure 1, five π-COs **O**s1~5 can be simultaneously produced in a single pot of non-equimolar direct C–H arylation coupling, wherein DPP was used in excess (1.5 eq.) to ensure the formation of π-COs end-capped by C–H bonds. Owing to the excessive amount of DPP (1.5 eq.), the chain growth will be stopped automatically after consuming the C–Br bonds of Cz, and meanwhile produce π-COs chains exclusively terminated by DPP units. This strategy of one pot reaction simultaneously affording five π-COs has distinct advantages with regard to the cost-effectiveness, labor- and energy-saving, and atomic economy. The repeating unit numbers of **O**s1~5 gradually increase, and progressively consist of 2, 3, 4, 5 and 6 DPP units, and 1, 2, 3, 4 and 5 Cz units, respectively. These π-COs can be smoothly separated by a single CC successively from the short-chain ones to the long-chain ones by gradually tuning the ratio of CH_2_Cl_2_/petroleum mixed eluent (Appendix A, see Appendix A). As a result, the isolated yields of **O**1, **O**2, **O**3, **O**4 and **O**5 are 21%, 16%, 13%, 12%, and 10% respectively, corresponding to a 72% total conversion of the starting DPP and Cz. In parallel, the parent polymer **P**1 was obtained via equimolar coupling between DPP and Cz in a yield of 83% based on CHCl_3_ fraction from Soxhlet extraction. All five π-COs and their parent polymer **P**1 possess good solubility in common organic solvents, such as CH_2_Cl_2_, CHCl_3_ and toluene, ensuring a solution-processability for device fabrication.

These new π-COs were characterized by NMR, Maldi-Tof MS and elemental analysis. Considering the analogous building blocks of **O**s1~5 and **P**1, we paid attention to their ^1^H NMR spectra of aromatic protons (Figure 1). As shown in Figure 1a, the homologous characters of **O**s1~5 and **P**1 can be directly reflected by the analogous chemical shifts in the aromatic region. The signals at 7.29–7.32 and 8.91–8.94 ppm having fixed integrals of 2 for each π-CO and **P**1are assigned to the protons on the terminal thiophene rings (H*_h_* and H*_b_*). The signals at 8.10–8.17 and 9.07–9.14 ppm are assigned to H*_c_* and H*_a_* on the backbone thiophene rings, both of which exhibit integrals of 2, 4, 6, 8 and 10, corresponding to the linear elongation of the oligomers from **O**1 to **O**5. The repeating unit numbers of **O**s1~5 can also be directly reflected by H*_a_*/H*_b_* integral ratios, i.e., the H*_a_*/H*_b_* ratios for **O**s1~5 are 1, 2, 3, 4 and 5, respectively, which exactly equal the repeat unit numbers of the π-COs. The H*_a_*/H*_b_* ratio of **P**1 is 34/1, and the degree of polymerization (DP) of **P**1 can also be inferred from this correlation. The equimolar polycondensation between DPP and Cz let the polymer chains have equal chance to be end-capped either by Cz or by DPP. Thus, the DP of **P**1 should be half (17) of the H*_a_*/H*_b_* ratio (34). Taking the end-capped building blocks into account, the average number of repeating units of **P**1 should be ~18. Thus, the average molecular weight (MW) of **P**1 estimated from ^1^H NMR integration should be 18 × 927 = 16,686, where 927 is the MW per repeat unit. The MWs of **O**s1~5 detected by Maldi-Tof MS are 1449.904, 2376.625, 3302.268, 4231.378 and 5161.625, which match well with the calculated values. The MWs of **O**s1~5 exhibit an interval value of 927 that equals the MW per repeat unit (Figure 1b), reinforcing the stepwise elongation from the shortest **O**1 to the longest **O**5.

The number average molecular weight (Mn) and polydispersity index (PDI) of **O**s1~5 and **P**1 were estimated by gel permeation chromatography (GPC) using polystyrene (PE) as standard and THF as eluent. The Mns are 2150, 3123, 4384, 5257, 6357 and 65,082, corresponding to PDIs of 1.05, 1.07, 1.07, 1.06, 1.07 and 1.60 for **O**s1~5 and **P**1, respectively. All π-COs exhibit PDI values around 1 with near mono-disperse distribution of Mn (Appendix A). The GPC Mns of π-COs exhibit overestimation compared to those from Maldi-tof MS, mainly owing to the rigid π-CO chains compared to the random-coil PE standard.

### 2.2. Opto-Electrochemical Property Study

Typically, the opto-electronic properties of π-conjugate materials can be finely tuned via introducing D-A architecture to modulate the electron push-pull effect. Here, the DPP-Cz-based D-A π-COs (Figure 1) with gradually elongated conjugations will be ideal models for a structure-property-performance correlation study. The opto-electrochemical properties of **O**s1~5 and **P**1 were investigated by UV-vis absorption and photoluminescence (PL) spectroscopies and a cyclic voltammetry (CV) test (Figure 2 and Table 1). As shown in Figure 2a,b, these five π-COs with A.A-D-A.A backbones exhibit broad light absorption ranging from 450 to 800 nm both in diluted chloroform (CF) solutions and solid state films. The structural evolution from **O**1 to **O**5 and **P**1 leads to gradual red-shifts in light absorption, implying the enhanced intramolecular charge transfer (ICT) with the elongation of π-conjugation. All π-COs and **P**1 exhibit similar spectral profiles involving two bands at 320–500 and 500–700 nm, which are due to the localized π-π* transition and intermolecular charge transfer associated with interchain π-π transition, respectively.

The maximum absorption peaks of **O**s1~5 and **P**1 in solutions (λ_max_^s^) are located at 598, 603, 645, 650, 652.5 and 658 nm, respectively. The red-shifts of **O**s2~5 and **P**1 compared to **O**1 are 5, 47, 52, 54.5 and 60 nm, respectively, as a result of the increase of π-conjugation length. Corresponding to the gradual red-shifts of light absorption, the colors of **O**s1~5 and **P**1 solutions evolved from purple, blue-purple, blue, blue-green to green (Figure 1). The extinction coefficients (ε) of **O**s1~5 solutions at λ_max_^s^ are 6.3 × 10^4^, 1.2 × 10^5^, 1.9 × 10^5^, 2.8 × 10^5^ and 3.8 × 10^5^ M^−1^·cm^−1^ respectively, showing a near linear-correlation with the numbers of the repeating units (i.e., DPP-Cz) involved. The π-COs and **P**1 have two absorption peaks at 550~590 nm and 605~660 nm, respectively. As shown in Figure 2b and Appendix A, the relative intensity of the peaks at longer wavelengths compared to the peak at shorter wavelengths increase gradually as the π-COs evolved from **O**1 to **O**5 and **P**1 due to the increased interchain interaction with the elongation of π-conjugations. The light absorption onsets (λ_onset_) of **O**s1~5 and **P**1 are 708, 725, 749, 756, 761 and 776 nm, respectively. Accordingly, the optical bandgaps (E_g_^opt^) of **O**s1~5 and **P**1 calculated from 1240/λ_onset_ are 1.75, 1.71, 1.66, 1.64, 1.63 and 1.60 eV, respectively.

The thin film of **O**s1~5 and **P**1 show a broadened absorption region with the λ_max_^f^ at 575, 614, 644, 638, 704.5 and 696 nm, respectively. Compared to solutions, the light absorption of thin films (λ_max_^f^) of the π-COs and **P**1 exhibit a distinct broadened shift and peaks at the ICT band (Figure 2b), indicating the strong intermolecular π-π stacking in the solid-state films. Owing to the aggregated solid state, the thin films of π-COs and **P**1 have two more pronounced vibration peaks with a broader spectrum compared to their corresponding solutions. The PL spectroscopy test (Figure 2c) was employed to reveal the reorganization energy between the ground and excited-state transition. The CF solutions of **O**s1~5 excited by 500, 520, 546, 547, and 548 nm gave emission peaks (λ_em_) at 646, 688, 692, 698 and 702 nm, respectively. The Stokes shifts (SS) of **O**s1~5 are 43, 43, 42, 43, and 44 nm, respectively.

The frontier molecular orbital (FMO) levels were evaluated by CV test. The highest occupied molecular orbital (HOMO) levels of π-COs and **P**1 can be estimated from CV curves (Appendix A). The corresponding lowest unoccupied molecular orbital (LUMO) levels were calculated from E_LUMO_ = E_HOMO_ + E_g_^opt^. The HOMO/LUMO levels of **O**s1~5 and **P**1 are estimated to be −5.47/−3.99, −5.41/−3.70, −5.39/−3.73, −5.37/−3.74, −5.35/−3.72 and −5.32/−3.71 eV, respectively (Figure 2d). The HOMO levels display an up-shift trend from **O**1 to **O**5, owing to the increasing ratios of electronic-donating Cz with the increase of π-conjugation lengths, i.e., the DPP/Cz ratios for **O**1, **O**2, **O**3, **O**4 and **O**5 are 2/1, 3/2, 4/3, 5/4 and 6/5 (Figure 1), respectively, which also accounts for the deepest LUMO level of **O**1.

### 2.3. BHJ OPV Performance Study

The OPV performances of the π-COs and **P**1 were studied by a device architecture of glass/ITO/PEDOT:PSS/BHJ layer/PFN-Br/Ag, wherein the BHJ layers consist of π-COs (or **P**1) and PC_70_BM as donor and acceptor, respectively. All BHJ layers were fabricated by spin-coating with π-COs/PC_70_BM mixed solution on unheated substrates. The current density-voltage (*J*–*V*) curves and the photovoltaic parameters of the BHJ devices are depicted in Figure 3 and Table 2. The power conversion efficiencies (PCE) π-COs increases with the increase of π-conjugation length, and the PCE of **O**5 is the closest to that of **P**1. Despite the fact that π-COs and **P**1 have low PCEs, the *J*_SC_ of **O**4 and **O**5 are very close to that of **P**1, which can be attributed to **O**4 and **O**5, which are close to the effective conjugation length. In addition, the fill factor (*FF*) of **O**s1~5 increases with the increase of chain length. The *FF* of **O**5 even exceeds that of **P**1. Although the π-COs decreases in *V*_OC_s with the increase of chain length, however, all π-COs exhibit higher *V*_OC_s compared to **P**1 (0.75 V), which can be attributed to a deeper HOMO level of the π-COs (Figure 2d) that have larger offsets with the LUMO of PC_70_BM.

## 3. Materials and Methods

### 3.1. Measurements and Reagents

Unless otherwise specified, all of the conventional chemicals were purchased from Energy Chemical. Cz and DPP were purchased from SunaTech Inc., Soochow. The anhydrous toluene was obtained by distillation after calcium hydride treatment. The ^1^H and ^13^C NMR spectra were obtained from the samples dissolved in CDCl_3_ using a Bruker 400 spectrometer (^1^H NMR 400 MHz and ^13^C NMR 101 MHz). MALDI-TOF MS were performed on a Bruker Auto flex II using 2,5-dihydroxybenzoicacid or α-cyano-4-hydroxycinnamicacid as the matrixes. Elemental analyses were conducted on a Flash EA 1112 elemental analyzer. The Mn, Mw and PDI of all oligomers and polymers were measured with the GPC (Viscotek TDA302 triple detector) using THF as the eluent. The UV-vis absorption spectra were measured by a Shimadzu UV-2450 spectrophotometer. Cyclic voltammetry (CV) was done on a CHI 661C electrochemical workstation with a Pt disk, Pt plate, and standard 10 calomel electrode (SCE) as the working electrode, counter electrode, and reference electrode, respectively, in a 0.1 mol L^−1^ tetrabutylammonium hexafluorophosphate (Bu_4_NPF_6_) CH_2_Cl_2_ solution.

### 3.2. Synthesis of Os1~5

DPP (209.60 mg, 0.399 mmol, 1.5 equiv), Cz (150 mg, 0.266 mmol, 1 equal), PivOH (8.15 mg, 30% mol), anhydrous Cs_2_CO_3_ (173.50 mg, 0.533 mmol), Pd_2_(dba)_3_ (3.70 mg, 1.5 mol%), and tris(o-methoxyphenyl) phosphine (2.90 mg, 3 mol%) were added into a Schlenk tube. The mixture in the Schlenk tube was evacuated and argon-filled three times. Next, 5 mL toluene was added into the tube via a syringe and the air in the Schlenk tube was removed by freeze-vacuum-thaw cycles three times. The argon-filled tube was then placed in an oil pot at 100 °C for 24 h. Removal of the toluene by rotary evaporator afforded the crude product, which was purified by CC on silica gel using a mixture of CH_2_Cl_2_ and hexane as an eluent (The volume ratio CH_2_Cl_2_/hexane gradually increased from 0.8 to 3) and successively gave **O**s1~5, **O**1 (58 mg, yield 21%), **O**2 (42 mg, yield 16%), **O**3 (35 mg, yield 13%), **O**4 (32 mg, yield 12%), **O**5 (26 mg, yield, 10%).

**O**1 (DPP-Cz-DPP): ^1^H NMR (400 MHz, CDCl_3_) δ 9.11 (dd, J = 16.9, 3.6 Hz), 8.93–8.88 (m), 8.16–8.07 (m), 7.86 (s), 7.70–7.54 (m), 7.30–7.27 (m), 4.64 (s), 4.18–4.02 (m), 2.48–2.25 (m), 2.08–1.86 (m), 1.31 (ddd, J = 67.7, 37.5, 27.1 Hz), 0.99–0.85 (m), 0.78 (t, J = 6.9 Hz).

MALDI-TOF MS (m/z): [M]^+^ calcd. For C_89_H_119_N_5_O_4_S_4_, 1451.188, found 1449.904.

Elemental analysis: calcd for: C_89_H_119_N_5_O_4_S_4_, C, 73.66; H, 8.27; N, 4.83%. Found: C, 73.54; H, 8.22; N, 4.79%.

**O**2 (DPP-(Cz-DPP)_2_): ^1^H NMR (400 MHz, CDCl_3_) δ 9.11 (s), 8.91 (dd, J = 3.8, 1.0 Hz), 8.13 (dd, J = 11.7, 8.4 Hz), 7.87 (s), 7.71–7.54 (m), 7.28 (dd, J = 5.0, 4.0 Hz), 4.7 –4.58 (m), 4.13 (dd, J = 23.3, 15.6 Hz), 2.35 (dd, J = 9.5, 4.5 Hz), 2.01 (dd, J = 8.9, 4.1 Hz), 1.44–1.10 (m), 1.00–0.85 (m), 0.79 (t, J = 6.9 Hz).

MALDI-TOF MS (m/z): [M]^+^ calcd. For C_148_H_198_N_8_O_6_S_6_, 2377.596, found 2376.625.

Elemental analysis: calcd for: C_148_H_198_N_8_O_6_S_6_ C, 74.76; H, 8.39; N, 4.71%. Found: C, 74.68; H, 8.32; N, 4.75%.

**O**3 (DPP-(Cz-DPP)_3_): ^1^H NMR (400 MHz, CDCl_3_) δ 9.09 (d, J = 14.8 Hz), 8.90 (d, J = 3.3 Hz), 8.12 (s), 7.87 (s), 7.63 (dd, J = 21.8, 16.8 Hz), 4.65 (s), 4.11 (d, J = 37.9 Hz), 2.36 (d, J = 7.4 Hz), 2.05 (s), 1.43–1.15 (m), 0.99–0.86 (m), 0.79 (t, J = 6.5 Hz).

MALDI-TOF MS (m/z): [M]^+^ calcd. For C_207_H_277_N_11_O_8_S_8_, 3304.003, found 3302.268.

Elemental analysis: calcd for: C_207_H_277_N_11_O_8_S_8_, 75.25; H, 8.45; N, 4.4.66%. Found: C, 75.28; H, 8.42; N, 4.69%.

**O**4 (DPP-(Cz-DPP)_4_): ^1^H NMR (400 MHz, CDCl_3_) δ 9.11 (d, J = 16.8 Hz, 8H), 8.94 (d, J = 3.1 Hz, 2H), 8.19–8.09 (m, 8H), 7.90 (s, 4H), 7.7–7.56 (m, 22H), 7.33–7.29 (m, 2H), 4.68 (s, 4H), 4.14 (d, J = 39.6 Hz, 20H), 2.39 (s, 8H), 1.99 (d, J = 55.5 Hz, 18H), 1.48–1.15 (m, 176H), 1.03–0.89 (m, 60H), 0.84–0.78 (m, 24H).

MALDI-TOF MS (m/z): [M]^+^ calcd. For C_266_H_356_N_14_O_10_S_10_, 4230.411, found 4231.378.

Elemental analysis: calcd for: C_266_H_356_N_14_O_10_S_10_, 75.52; H, 8.48; N, 4.64%. Found: C, 75.48; H, 8.45; N, 4.69%.

**O**5 (DPP-(Cz-DPP)_5_):^1^H NMR (400 MHz, CDCl_3_) δ 9.13 (d, J = 16.1 Hz, 10H), 8.94 (d, J = 3.1 Hz, 2H), 8.13 (d, J = 8.4 Hz, 10H), 7.90 (s, 5H), 7.72–7.59 (m, 27H), 7.32–7.29 (m, 2H), 4.67 (s, 5H), 4.14 (d, J = 39.0 Hz, 24H), 2.39 (s, 10H), 2.06 (s, 24H), 1.45–1.16 (m, 216H), 1.04–0.91 (m, 72H), 0.82 (dd, J = 6.7, 4.4 Hz, 30H).MALDI-TOF MS (m/z): [M]^+^ calcd. For C_325_H_435_N_17_O_12_S_12_, 5156.818, found 5156.625.

Elemental analysis: calcd for: C_325_H_435_N_17_O_12_S_12_, 75.70; H, 8.50; N, 4.62%. Found: C, 75.68; H, 8.47; N, 4.64%.

### 3.3. Synthesis of P1 ((Cz-DPP)_n_)

DPP (139.71 mg, 1 equiv) and Cz (150 mg, 1 equal) were used as reactants. Other additives are the same as the synthesis of π-COs. After polymerization for 48 h, the crude product was purified by precipitation in methanol, filtered, and washed on Soxhlet with methanol acetone, hexanes, and chloroform successively. The chloroform fraction was condensed under reduced pressure, and obtained the polymer **P**1: (238 mg, yield 83%).

^1^H NMR (400 MHz, CDCl_3_) δ 9.13 (d, J = 14.4 Hz, 2H), 8.93 (s, 1H), 4.66 (s, 1H), 4.11 (d, J = 69.4 Hz, 4H).

### 3.4. BHJ Device Fabrication

The conventional glass/ITO/PEDOT:PSS/BHJ layer/PFN-Br/Ag structure was employed to fabricate the OPV devices. The indium tin oxide (ITO) substrates were cleaned sequentially with deionized water and isopropyl alcohol by ultrasonication, and then dried in an oven at 60 °C overnight. The dried ITO substrates were treated with oxygen plasma for 5 min and then coated with PEDOT:PSS at 3000 rpm for 60 s. The film was annealed on a hot dish at 150 °C in air for 15 min to obtain a thickness of about 40 nm. The substrates were then transferred to a N2-protected glove box. The active layer was obtained by spin-coating a chlorobenzene solution containing D: A (1:1) and 0.5% (*v/v*) 8-diiodooctane (DIO) additive at a total concentration of 20 mg mL^−1^. The optimum film thickness of about 100 nm was obtained by controlling the rotation speed of about 2500 rpm. The substrate coated with the active layer was then placed on a hot plate and thermally annealed for 10 min. Subsequently, about 6 nm of PFN-Br was spin-coated onto the active layer as a cathode interfacial layer. Finally, 100 nm of silver was thermally deposited on top of the interface by a mask in a vacuum chamber at 1 × 10^−7^ Torr pressure. The effective area of the device is 0.04 cm^2^.

## 4. Conclusions

A series of DPP-Cz based π-COs **O**s1~5 with gradually increasing π-conjugation lengths were designed and facilely synthesized via a one-pot direct C–H arylation for opto-electronic property and OPV performance study. All π-COs involve the same building blocks (i.e., DPP and Cz), but have different conjugation lengths. The **O**s1~5 and their parent polymer **P**1 were employed as donor materials to pair with PC_71_BM for BHJ OPV device study. The results obtained here demonstrate that D-A backboned π-COs with progressively increased conjugation chain lengths via one pot C-H/C-Br coupling is a promising way to design and synthesize π-COs for BHJ OPV device applications and expand the horizons of π-functional materials.

## Data Availability

Not applicable.

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
