# Peer review of "Carbazole and Diketopyrrolopyrrole-Based D-A π-Conjugated Oligomers Accessed via Direct C–H Arylation for Opto-Electronic Property and Performance Study"

_molecules, 2022, doi:10.3390/molecules27249031_

Round 1

Reviewer 1 Report

Liu et al. synthesized a series of carbazole and diketopyrrolopyrrole-based conjugated oligomers via the direct C–H Arylation, and systematically investigated the molecular weight on their optical, electrical and photovoltaic properties. Overall, the manuscript is well-organized, and the conclusion is strongly supported by the data, thus the manuscript can be accepted by addressing the below minors:

1.      Why the LUMO of O1 is much deeper than other oligomers and P1?

2.      The CV of the oligomers and P1 were measured in solution or film state?

3.      Only the extinction coefficients in solution were measured, how about the extinction coefficients of the oligomer and polymer in film state (DOI: 10.1002/aenm.201300046)?

4. A-D-A type oligomer based small molecular donors or acceptors have been systematically investigated in the past 12 years (DOI: 10.1021/ar400088c; 10.1007/s11426-022-1264-y), and the pioneering works should be included.

5. As a suggestion, all the authors should use their email with affiliation.

Author Response

Response to Reviewer 1 Comments

Liu et al. synthesized a series of carbazole and diketopyrrolopyrrole-based conjugated oligomers via the direct C–H Arylation, and systematically investigated the molecular weight on their optical, electrical and photovoltaic properties. Overall, the manuscript is well-organized, and the conclusion is strongly supported by the data, thus the manuscript can be accepted by addressing the below minors:

We appreciate the positive comments made by the reviewer: “ …Overall, the manuscript is well-organized, and the conclusion is strongly supported by the data, thus the manuscript can be accepted by addressing the below minors …”.

Point 1: Why the LUMO of O1 is much deeper than other oligomers and P1?

Response 1: Thanks, this is really a good question! We also noted that O1 shows the deepest LUMO. Typically, the LUMO level of a molecule is governed by its electon accepting moieties. The ratios of electron-accepting DPP to electron-donating Cz (i.e., the DPP/Cz ratios) for O1, O2, O3, O4 and O5 are 2/1, 3/2, 4/3, 5/4 and 6/5, respectively. Among them, O1 has the highest ratio of electron-accepting DPP, which naturally endows it’s the deepest LUMO level. In this revised version, this explaination have been supplied (please kindly find it in page 6).

Point 2: The CV of the oligomers and P1 were measured in solution or film state?

Response 2: Thanks for comment. The CV of π-COs and P1 were measured in CH2Cl2 solutions. In the experimental section of this revised version, we have detailed the test method of CV (please kindly find it in page 7).

Point 3: Only the extinction coefficients in solution were measured, how about the extinction coefficients of the oligomer and polymer in film state (DOI: 10.1002/aenm.201300046)?

Response 3: This is a good suggestion. In our future work, we will surely pay attention to this issue.

Point 4: A-D-A type oligomer based small molecular donors or acceptors have been systematically investigated in the past 12 years (DOI: 10.1021/ar400088c; 10.1007/s11426-022-1264-y), and the pioneering works should be included.

Response 4: Thanks for nice suggestion. These pioneering works have been involvled in this revised version.

Point 5: As a suggestion, all the authors should use their email with affiliation.

Response 5: Thanks for suggestion. Lingwei Feng's institutional email address has been supplied in this revised version. Xiafeng Zhang has graduated, and she has no institutional email address.

Reviewer 2 Report

Liu et al reported a series of DPP-Cz based π-COs with gradually increasing π-conjugation lengths that were synthesized via a one-pot non-equimolar direct C–H arylation coupling, wherein DPP was used in excess (1.5 eq.) to ensure the formation of π-COs end-capped by C–H bonds of DPP. The structure-property-performance correlations of these π-COs with gradually increasing lengths were systematically studied by opto-electrochemical tests and OPV devices. This is an interesting and impressive study that develops a highly effective synthetic protocol toward π-COs materials, especially for long-chain ones. The manuscript is also well-written and well lay-out. I would like to recommend it for publication after proper revision.

[1] The languange still can be improved if proofreading is done.

[2]  Please check again the figure quality overall.

[3] In view of recent progress of PV field, could author give more discussion of carbazole unit's potential in further boosting efficiency?

Author Response

Response to Reviewer 2 Comments

Liu et al reported a series of DPP-Cz based π-COs with gradually increasing π-conjugation lengths that were synthesized via a one-pot non-equimolar direct C–H arylation coupling, wherein DPP was used in excess (1.5 eq.) to ensure the formation of π-COs end-capped by C–H bonds of DPP. The structure-property-performance correlations of these π-COs with gradually increasing lengths were systematically studied by opto-electrochemical tests and OPV devices. This is an interesting and impressive study that develops a highly effective synthetic protocol toward π-COs materials, especially for long-chain ones. The manuscript is also well-written and well lay-out. I would like to recommend it for publication after proper revision.

We appreciate the positive comments made by the reviewer: “ …This is an interesting and impressive study that develops a highly effective synthetic protocol toward π-COs materials, especially for long-chain ones. The manuscript is also well-written and well lay-out. I would like to recommend it for publication …”.

Point 1: The languange still can be improved if proofreading is done.

Response 1: Thanks for nice suggestion. We have carefully re-checked the whole manuscript to improve to the language quality and make it more readable.

Point 2: Please check again the figure quality overall.

Response 2: Thanks very much! The figure quality and resolution have been improved in this revised version.

Point 3: In view of recent progress of PV field, could author give more discussion of carbazole unit's potential in further boosting efficiency?

Response 3: This is a good suggestion. In this revised version, the discussion on the potential of carbazole unit in OPV fielf have been supplied (please kindly find it in page 3).